# A Cost-Efficient High-Speed VLSI Architecture for Spiking Convolutional Neural Network Inference Using Time-Step Binary Spike Maps

**DOI:** 10.3390/s21186006

**Published:** 2021-09-08

**Authors:** Ling Zhang, Jing Yang, Cong Shi, Yingcheng Lin, Wei He, Xichuan Zhou, Xu Yang, Liyuan Liu, Nanjian Wu

**Affiliations:** 1School of Microelectronics and Communication Engineering, Chongqing University, Chongqing 400044, China; zhangling1993@cqu.edu.cn (L.Z.); yang_jing@cqu.edu.cn (J.Y.); linyc@cqu.edu.cn (Y.L.); hewei007@cqu.edu.cn (W.H.); zxc@cqu.edu.cn (X.Z.); 2Key Laboratory of Dependable Service Computing in Cyber Physical Society, Ministry of Education, Chongqing University, Chongqing 400044, China; 3State Key Laboratory of Superlattices and Microstructures, Institute of Semiconductors, Chinese Academy of Sciences, Beijing 100083, China; yangxu@semi.ac.cn (X.Y.); liuly@semi.ac.cn (L.L.); nanjian@red.semi.ac.cn (N.W.)

**Keywords:** neuromorphic computing, spiking convolutional neural networks, SNN hardware, VLSI implementation, pixel stream processing

## Abstract

Neuromorphic hardware systems have been gaining ever-increasing focus in many embedded applications as they use a brain-inspired, energy-efficient spiking neural network (SNN) model that closely mimics the human cortex mechanism by communicating and processing sensory information via spatiotemporally sparse spikes. In this paper, we fully leverage the characteristics of spiking convolution neural network (SCNN), and propose a scalable, cost-efficient, and high-speed VLSI architecture to accelerate deep SCNN inference for real-time low-cost embedded scenarios. We leverage the snapshot of binary spike maps at each time-step, to decompose the SCNN operations into a series of regular and simple time-step CNN-like processing to reduce hardware resource consumption. Moreover, our hardware architecture achieves high throughput by employing a pixel stream processing mechanism and fine-grained data pipelines. Our Zynq-7045 FPGA prototype reached a high processing speed of 1250 frames/s and high recognition accuracies on the MNIST and Fashion-MNIST image datasets, demonstrating the plausibility of our SCNN hardware architecture for many embedded applications.

## 1. Introduction

Neuromorphic computing has attracted ever-increasing interest in the past ten years. The spiking neural network (SNN) closely mimics the operational mechanism in the human brain cortex, where information is encoded, communicated, and processed via very sparse electrical pulses (i.e., spikes) among neurons, ensuring high-energy efficiency in cognitive tasks [1]. SNNs exhibits a radical computing paradigm shift from their traditional artificial neural network (ANN) counterparts. ANNs employ dense computations and all of the neurons have to participate in an inference, while SNNs leverage temporally sparse spike trains and may activate only a small portion of neurons during the inference. Another difference is that SNNs require a time dimension to evolve with the temporal spike trains. Therefore, general-purpose computers such as the Central Processing Unit (CPU) and the Graphics Processing Unit (GPU) are incompetent in deploying brain-inspired SNN models, as those von Neumann machines are oriented for dense numerical calculations rather than sparse temporal spike processing.

To fully exploit the computational and energy efficiency of SNNs, various dedicate neuromorphic chips and hardware systems have recently been designed [2,3,4,5,6,7,8,9,10,11,12,13,14]. These VLSI chips support various spiking neuron models at different levels of biological fidelity and computational complexity, and generally adopt scalable routing schemes including crossbars and network-on-chip (NoC) infrastructures towards large-scale or even brain-scale multichip systems. Each spike is delivered among neural computing nodes as an address-event representation (AER) data packet, affiliated with the source and/or destination neuron identities (addresses), and, optionally, the time stamps indicating when this spike has been issued. Arbitrary network connections can be implemented under such point-to-point AER protocol. As a result, the neuromorphic chips have provided a real-time, highly flexible hardware framework to simulate and investigate various structures, functions, and operations in different cortical regions for biomedical and neuroscientific research.

On the other side, for real-world applications, especially for visual cognitive tasks, the most plausible and widely used network topology is the feedforward convolutional structure, which is composed of stacked convolutional (CONV) layers for hierarchical feature extraction, and fully connected (FC) layers for feature classification. Such network structure has gained astonishing high accuracies in recognizing complicated visual objects in the ANN domain [15,16,17]. Likewise, the spiking convolution neural network (spiking CNN, or SCNN) is supposed to achieve relatively higher recognition accuracies in neuromorphic visual processing [18,19,20,21,22,23,24]. These SCNN models can be trained off-line before they are loaded to neuromorphic chips for object inference [4,11]. However, those reported neuromorphic chips are not adequate architectures for SCNNs, as they cannot utilize the following structural regularity of the CONV layers during inference: (1) only feedforward connections between successive layers are needed, (2) each neuron is only connected to a few neurons in a small neighborhood in its precedent layer (i.e., the receptive field of the neuron), (3) the set of convolutional kernel weights are shared across all the neurons in the same channel of one layer. Instead, those neuromorphic chips must replicate the shared convolutional weights in the synapse memory for all neurons, and use the redundant crossbar to designate the feedforward, small-neighborhood connections, where most crossbar nodes are disconnected, and hardware resources are wasted. Moreover, as the network goes deeper, there is an increasing probability of spike collisions on their NoC routers, and the processing latency or inference accuracy of the SCNN may deteriorate.

To overcome these problems, some neuromorphic hardware specialized for deep SCNNs are designed. Configurable event-driven convolution chips are proposed in [25,26,27] to support multiple convolution kernels and different kernel sizes. Two CONV layers are built and cascaded in [25] for fast card symbol recognition, which are then extended to four CONV layers with subsampling in [26]. More efficient memory access is realized on an FPGA prototype in [27], by using a novel memory arbiter. A 28 nm SCNN processor is fabricated in [28], holding one CONV layer of 10 kernels for feature extraction, one pooling layer for dimension reduction, and two FC layers for feature classification. A systolic SCNN inference engine is proposed in [29], and two CONV layers with one FC layer are instantiated. A SNN inference engine called SIES is proposed in [30]and uses 2-D systolic array to accelerate the computation of the CONV layer.

The main contributions of our work include: (1) this paper proposes a scalable, high-speed and low-cost neuromorphic VLSI architecture for SCNN inference in real-time and resource-constrained application scenarios (e.g., portable or mobile platforms, edge-computing systems, internet-of-things devices, etc.); (2) we leverage the snapshot of binary spike maps at each time-step along with the spike-map pixel stream processing pipeline to maximize spike throughput, while minimizing the computation and storage consumptions of hardware resources; (3) this architecture was prototyped on an FPGA platform with different SCNN depth configurations. Up to 1250 frames/s throughput on the 28 × 28 MNIST images were obtained under a 100 MHz clock frequency. 

The rest of this paper is organized as follows. Section 2 briefly reviews the spiking neuron model behaviors and the SCNN operation principle. Section 3 proposes our concept of time-step binary spike map, as well as its regular, simple, and high-performance processing mechanism. The SCNN hardware design details are described in Section 4. Section 5 evaluates our architecture on FPGA prototypes and compares our work with others. Finally, a conclusion is drawn in Section 6.

## 2. Background

### 2.1. Biological Spiking Neuron Model

Current neuromorphic studies have led to the development of numerous models [19,27,31] to simulate biological neurons at different levels of abstraction. Among them, the leaky integrate-and-fire (LIF) model is particularly popular due to its low computational complexity and sufficient biological fidelity [19,27]. The dynamical behavior of the LIF spiking neuron in the discrete time domain can be described as: (1)$$V_{m}(t)=V_{m}(t-1)(1-\tau_{m}^{-1})+\sum_{i}w_{i}s_{i}(t)$$
where *t* is the discrete time-step, *V*_m_(*t*) is the neuronal membrane potential, *τ*_m_ is the leakage time constant, *w_i_* is the weight of the neuron’s *i*-th synapse, and *s_i_*(*t*) is the input (pre-synaptic) spike train at that synapse: *s_i_*(*t*) is 1 if synapse *i* receives a spike at time *t*, and 0 otherwise. As shown in Figure 1, The LIF neuron continuously integrates input spikes onto its membrane potential via the synaptic weights, and exponentially leaks at the moments when no input spike occurs. Once its membrane potential crosses a predefined threshold *V*_th_, the neuron fires an output (post-synaptic) spike and immediately resets *V*(*t*) to a resting level (usually zero, as in this work). Such leaky integrate-and-fire process repeats until the end of input spike trains. 

### 2.2. Spiking Convolution Neural Network 

SCNN is a type of feedforward SNN consisting of a stack of CONV layers for hierarchical feature extraction, followed by one or several FC layers as the feature classifier, as depicted in Figure 2 [18,24,32]. A CONV layer is composed of multiple channels of neuronal maps holding 2-D arrays of spiking neurons. Each spiking neuron in one map is connected to a few neurons in a small spatial neighborhood (i.e., the receptive field, equaling to the convolutional kernel size) across all channel maps in its preceding layer. And all of the neurons in the map share the same set of convolutional kernels. Therefore, the LIF spiking neuron in a CONV layer can be particularized as: (2)$$V_{m}(x,y,c_{\mathrm{out}},t)=V_{m}(x,y,c_{\mathrm{out}},t-1)(1-\tau_{m}^{-1})+\sum_{c_{\mathrm{in}}}\sum_{p,q}w(p,q,c_{\mathrm{in}},c_{\mathrm{out}})s(x+p,y+q,c_{\mathrm{in}},t)$$
where *x*, *y* are the spatial locations of the spiking neuron on the map, *c*_out_ and *c*_in_ represent the channel indices of this CONV layer and its input layer, respectively. *w* is a 4-D tensor of the convolutional kernel weights, and *p*, *q* are spatial coordinates within the kernels. *s* is the spike trains from the input channels. 

In conventional CNNs, a pooling layer may exist between successive CONV layers for map dimension reduction. However, such pooling operations are much more sophisticated for hardware SCNN implementations [33]. An alternative way to reduce map dimension is to use CONV layers with a stride larger than 1. A CONV layer with a stride of *d* means that the convolutional kernels are placed only on every *d*-th row and every *d*-th column on the input map during the convolution process, as illustrated in the lower part of Figure 2. It is equivalent to performing a normal convolution (i.e., with a stride of 1), but only picking out the resultant pixels on every *d*-th row and every *d*-th column to constitute the output map.

To apply the SCNN to object recognitions, the object images must be encoded into spike patterns. In the commonly used rate coding scheme, each pixel of the image is converted to a spike train with an average spike frequency in proportional to the pixel intensity [34]. Each output neuron in the final FC layer is assigned to one object category. The object category who has the most output spikes is regarded as the classified result [35,36].

## 3. Proposed Spike Map Stream Processing Mechanism

In most previous SCNN hardware implementations, the spikes are delivered among neurons in the AER data format [37]. The AER package for one spike includes the input location or neuron index (i.e., address) where the spike is generated from, and optionally the polarity and timestamp of the spike. Therefore, a large field of multiple bits is needed to express the originally simple single-bit spike event, and complicated circuits are required to route and process such AER packages [3,4,25,26,27].

To reduce hardware complexity for mobile and edge systems, one solution is to still treat the spikes in a single-bit format, and the spike source address should be implicitly indicated in this format. For this purpose, we propose the concept of *the time-step binary spike map*, along with high-performance processing techniques for such binary maps. At any discrete time-step t in the digitalized SCNN, the output spikes of the neurons in one channel of the CONV layer can be snapshot in the form of a binary spike map (image), where the map pixels are 1 and 0 on the locations of firing and non-firing neurons at this time-step, respectively, as illustrated in Figure 3. The FC layer can be treated as a special CONV layer with 1 × 1 maps and 1 × 1 convolutional kernel size. Based on such concept of time-step binary spike map, we can now leverage regular and much simpler computing techniques for conventional non-spiking CNN with 1-bit binary feature maps rather than using complicated AER processing mechanisms. More specifically, within each single time-step t, the SCNN evaluation is equivalent to the CNN-like processing:(3)$$B(x,y,c_{L})=H(\sum_{c_{L-1}}\sum_{p,q}w(p,q,c_{L-1},c_{L})B(x+p,y+q,c_{L-1})+bias)$$
where *B* is the snapshot time-step binary spike map, *c_L_* indicates the channel index of layer *L,* and *w* is the convolutional kernel weights. The Heaviside function *H*(*x*) is 1 or 0, if *x* ≥ 0 or *x* < 0, respectively. It acts as a nonlinear transform function in the CNN-like processing. For each neuron, the *bias* term in Equation (3) is their respective membrane potential *V*_m_ at the previous time-step scaled by the leakage factor (1 − 1/τm), minus the preset firing threshold *V*_th_: *bias* = *V*_m_(1 − 1/*τ*_m_) − *V*_th_. The variable of time *t* is omitted in Equation (3), as it describes operations in the context of a single time-step. As a result, executing Equation (2) for SCNN inference along a time window 0 ≤ *t* < *T* can be decomposed into a series of regular and simple time-step CNN-like processing on the binary spike maps by using Equation (3).

Since the time-step CNN-like processing uses binary spike maps, the processing circuit consumptions can be reduced, and the processing performance can be improved. In this work, we propose a computationally efficient cost-effective binary spike map pixel stream processing mechanism for high-speed low-cost hardware implementation, as shown in Figure 4. For one particular layer at a certain time-step, the input binary spike map pixels from all the channels of its preceding layer are streamed in a pixel-serial channel-parallel manner. Only the latest rows and columns of the input pixel streams are buffered, producing parallel output streams of binary spike map pixels in this layer. Such stream-triggered processing propagates all of the layers until the final output streams of current time-step appear at the output layer. The advantages of employing such time-step spike map pixel stream processing mechanism for hardware implementation are threefold. (1) *Small memory footprint*: the spike maps of all layers are computed on the fly along the streams, without the need to store the whole map data of any layer. For each layer, only a very small portion (depending on the kernel size) of the latest rows and columns of each input binary map need to be buffered and updated to produce the currently desired output map pixels. Reducing memory consumption as much as possible is quite critical for on-chip implementation of large-scale SCNNs. (2) *High pixel throughput*: The pixel stream of each layer can be computed in a channel-parallel pixel-pipeline manner to achieve a high processing throughput up to 1 pixel/channel per-clock cycle, as will be revealed in the next section. (3) *Reduced computing resources*: in the map pixel stream processing pipeline, all the binary pixels in one channel are serially handled, thus a small pixel processing circuit can be reused to significantly save computing resources, with simple and fast adders instead of expensive hardware multipliers to complete the convolving operations.

## 4. VLSI Architecture

### 4.1. Architecture Overview

Figure 5 shows the proposed neuromorphic VLSI architecture for SCNN inference utilizing time-step binary spike maps. This architecture consists of an input spike map generator, *J* groups of CONV modules, *K* FC modules, pixel stream row buffers between CONV module groups, an output spike counter, and weight registers for CONV/FC layers. At every time step, the spike map generator snapshots the input AER stream as a binary spike map and sends the map pixels serially as a pixel stream to CONV modules. Each CONV module group corresponds to one CONV layer and one CONV module in such a group performs the CNN-like 3 × 3 pixel-stream-based convolutions for one channel in that CONV layer, and each FC module performs the computations needed for one FC layer, where one FC unit in the FC module corresponds to one spiking neuron in that FC layer. The spike counter calculates the fired spikes from the last FC module during all the inference time-steps 0 ≤ *t* < *T* of one input sample presentation, and it determines the final classification result. The circuit design details of the key computational blocks (i.e., the CONV and FC modules) in the architecture are further described below.

### 4.2. CONV Module Circuit

Figure 6 shows the computing circuit of the CONV module for evaluating Equation (3) on the pixel streams of binary spike maps, with 3 × 3 convolutional kernel sizes. The CONV computing datapath contains three components. The first component is a high-speed pipelined adder tree calculating the first term of the argument in the *H*() function in Equation (3), i.e., the sum of the products of kernel weights w and corresponding input spike map pixels B from all the previous layer’s channels. To produce one output spike map pixel, the adder tree simultaneously takes on the 3 × 3 weights (from the weight registers at the bottom of Figure 5) and 9 corresponding input pixels (fed by the pixel row buffers, as will be described in the next subsection) in each input channel at a time, in a channel-parallel fashion as shown in Figure 6, and then sums up the partial products from all of the input channels to get the whole product result. Since the adder tree works in a pipeline manner with each pipeline occupying 1 clock cycle, the whole adder tree can reach a high throughput up to 1 spike map pixel per clock cycle, or equivalently, 100 M pixel/s under a typical 100 MHz chip clock frequency. Note that since the spike map pixels are binary, the adder tree adopts simple bitwise AND gates rather than expensive hardware multipliers to obtain each *wB* product term. 

The second computing component of the CONV module adds the *bias* term in Equation (3) to the intermediate result from the first computing component, as shown in Figure 6. To reduce resource consumption and speedup calculation, the constant (1 − 1/*τ*_m_) is precalculated and stored in the parameter registers. Besides, the adding-bias component has an orthogonal datapath (from top to bottom in the middle of Figure 6) to necessarily update the neuronal membrane potentials stored in the *V*_m_ memory at each time step. Fortunately, this auxiliary datapath requires no extra computing logics as it can fully leverage the intermediate result during the bias addition operation, as illustrated in Figure 6.

The last computing component in the CONV module evaluates the Heaviside function *H*() in Equation (3), and gates the resultant output binary pixel stream to the next layer to realize striding convolution control. According to its definition, the *H*() block simply picks up the inversed sign bit of the result from the bias-adding component as the generated binary output map pixels. Finally, the stride controller only passes the output pixels at those row and column locations visited by the stride. For instance, in a convolution with a stride of 2, only those output pixels at both even rows and even columns can reach the next layer. The row and column counters are not drawn in Figure 6 for clarity reasons. Besides, the circuits of the second and third components in Figure 6 are also extremely simple and their operations for one output pixel can complete in only one clock cycle. Hence, they are further pipelined with the pipeline adder tree component. Therefore, the whole CONV module can run at a very high processing speed while consuming a small amount of hardware resources.

### 4.3. Pixel Row Buffer

The pixel row buffer is shown in Figure 7. It consists of two small 1-bit width memory pieces and 6 1-bit registers. One memory piece stores N-2 pixels, where N represents the horizontal size of its input spike map. The input pixel stream from its corresponding CONV module is buffered in a snake way, enabling 3 × 3 pixels (i.e., B[0] ~ B[8] in Figure 7) to be simultaneously provided to the 3 × 3 CONV module group belonging to the next layer at each clock cycle, which maximizes the system data throughput. Note that if the next layer is an FC layer, the row buffers are not needed. Instead, the pixel streams from all CONV channels are directly fed to each FC unit in the FC module in a pixel-serial channel-parallel fashion.

### 4.4. FC Unit Circuit

The FC layer unit circuit shown in Figure 8 performs the operation of one spiking neuron in the FC layer. Since the spiking neuron behaviors are identical in both CONV and FC layers in the SCNN model, the FC unit circuit is quite similar to that of the CONV module in Figure 6, executing in a pipelined high-throughput manner and also containing three components: a pipelined adder tree, an *adding-bias* block, and a Heaviside function evaluator. However, there are two differences between the circuits of the COVN modules and the FC units. The obvious difference is that the FC unit needs no stride control, and the output pixel stream from each FC unit contains only one pixel. Another difference is that the adder tree in the FC unit receives *C* (instead of the constant of 9 in the CONV module) input pixels and corresponding FC weights. If its preceding layer is an FC layer, *C* represents the number of neurons of the preceding FC layer (i.e., the number of FC units in its upstream FC module), and the sum of *wB* products in Equation (3) are directly sent to the *adding-bias* block once obtained. Otherwise, if the preceding layer is a CONV layer, *C* represents the number of the input CONV channels, and the adder tree must process the input pixel streams from the preceding CONV module group in a pixel-serial channel-parallel manner and buffers intermediate sum result in an accumulator register (*Acc.* in Figure 8), before the summation over the input convolution map pixels is completed.

## 5. Experimental Results

### 5.1. FPGA Prototype

In this work, we used the SystemVerilog to describe the proposed SCNN hardware architecture and the processing circuits prototyped on a Xilinx Zynq-7045 FPGA chip. The whole evaluation system of the prototype is shown in Figure 9. An on-chip hardware IP core of ARM processor was used to communicate the SCNN prototype and a host computer via a 1000 Mbps Ethernet link. To obtain the SNN weights, we trained equivalent CNN model structures in Pytorch in an offline manner, converted the CNN models into the desired SNN models using the method in [18], and finally downloaded the learned weights to the FPGA prototype via the Ethernet. We adopted four SCNN model configurations with different numbers of CONV and FC layers, as listed in Table 1, and the corresponding FPGA resource and power consumption (estimated by the Xilinx Vivado 2018.3 tool) for prototyping each of the four SNN models are given in Table 2. All of the four FPGA prototypes were running under a clock frequency of 100 MHz. Note that the on-chip hardware IP core of ARM processor and the PC debugging software in Figure 9 were only used for evaluation and they are not part of the prototype, thus their resource and power consumptions should not be counted in Table 2. Every test image was encoded into spike trains of 100 time-steps using the aforementioned rate coding method in our PC debugging software. These spike trains were recorded in the AER format and then downloaded to the FPGA prototype via the Ethernet. The image classification results were read back to the host computer for display and accuracy evaluation.

We used the MNIST image dataset [38] and the more challenging Fashion-MNIST dataset [39] to test our prototype, as shown in Figure 10. Each dataset contains 70,000 28 × 28 grayscale images belonging to 10 object categories, of which 60,000 are for offline SCNN training and the others for inference testing. We employed the method in [18] to train our SCNN in an offline manner on the host computer. For each SCNN model configuration, we first trained an equivalent non-spiking CNN using the standard error back-propagation algorithm, and then converted it to the target SCNN model following the routine in [18]. Finally, the trained SCNN weights were downloaded to the weight registers in the FPGA prototype for object category inference. Thanks to the proposed spike map pixel stream processing mechanism and the pipeline technique, the SCNN hardware processing throughput kept nearly constant regardless of the depth of SCNN layers. The prototype inference speed on the MNIST and Fashion-MINIST images for the SCNN model configurations in Table 1 were all as high as around 1250 frame/s (fps), under the 100 MHz clock frequency and with 100 time-steps for each image. The SCNN inference accuracies are illustrated in Figure 11. The recognition accuracy gap between the two datasets is due to the reason that the Fashion-MNIST object images from the real-world are much more complicated to classify than the handwritten digits in the MNIST dataset. We have found that compared to the 2C2F SCNN model, the 3C1F configuration has the same depth of 4 layers except that 1 FC layer is replaced by a CONV layer. However, the latter configuration achieved obviously high recognition accuracies on the MNIST and Fashion-MNIST images, while showing comparable hardware resource and power consumptions (Table 2) at almost the same high inference speed of about 1250 fps. This demonstrates the importance of the scalability to deeper CONV layers in the proposed SCNN hardware architecture.

### 5.2. Comparsion and Discussion

Table 3 compares our FPGA implementation with recent state-of-the-art softwarebased SNN/SCNN implementations regarding processing speed and the recognition accuracy. On the whole testing subset of 10000 MNIST images, our hardware implementation achieved processing accelerations of 1103×, 998×, and 638×, compared to the software works in [19,40,41], respectively, and is thus very suitable for real-time embedded applications. The work in [41] obtained the highest recognition accuracy using a more advanced CNN-to-SCNN training methodology. In the future, we can also utilize such technique to finish the offline SCNN training procedure and downloaded the learned weights to our hardware system to achieve similar high accuracy.

Table 4 compares our work with other recent SNN/SCNN hardware implementations. It demonstrates that our work achieves high processing speed, high recognition accuracy at a acceptably moderate resource and power consumption level. Although the customized neuromorphic ASIC chips exhibits higher resource and energy efficiencies as they occupy small chip area and consumes much lower power, they suffer from long design and verification cycle, high fabrication cost and less flexibility for chip function upgrade, when compared to the FPGA-based implementations. The work in [26] gains a little higher accuracy than ours, as they benchmarked their design on the much simpler Poker-DVS dataset which contains only 4 categories. The work in [33] also surpasses our work in terms of the MNIST recognition accuracy, but at a much lower frame rate of only 164 fps. The work in [30] achieved the highest accuracy among all the works, yet without reporting their power consumption and frame rate.

The architecture proposed in this paper allows for implementation of large-scale SCNN and presents competitive results in terms of the recognition accuracy and frame rate. It levearages the snapshot of binary spike maps at each time-step along with the spike-map pixel stream processing pipeline to improve the hardware throughput up to 1250 frame/s on MNIST and Fashion-MNIST image recognition tasks, while only consuming a few memory resources, a few multipliers, and a moderate amount of other logic cells on the FPGA prototype, as demonstrated in Table 2. Indeed, comapred to other SNN hardware implementations, listed in Table 4, that mainly utilize the NoC infrustructure and AER-based communications for arbitrary SNN model structures, our proposed architecture and circuits are specicially designed and optimized for the feedforward multi-layer SCNN models, which have hitherto the most widespread application in various visual recognition tasks, and thus achieve the highest frame rate on classifying the 28 × 28 resolution images.

## 6. Conclusions

In this paper, we propose a scalable, cost-efficient, and high-speed VLSI neuromorphic architecture for SCNN inferencing, which leverages the snapshot of binary spike maps at each time-step, to decompose the SCNN operations into a series of regular and simple time-step CNN-like processing. Our system architecture achieves high pixel throughput by employing the pixel stream processing mechanism, parallel processing arrays of CONV modules and FC units, and fine-grained pipelines. An FPGA prototype of the proposed hardware architecture was implemented on the Xilinx Zynq-7045 chip and reached a high processing speed of 1250 fps on the MNIST and Fashion- MNIST image datasets under a 100 MHz clock frequency. Our prototype achieved 97.3% and 83.3% recognition accuracies on the MNIST and Fashion-MNIST datasets using an SCNN model with 3 CONV layers and 1 FC layer, while moderately consuming 1.241W power. Our neuromorphic hardware system exhibits high plausibility for versatile embedded, mobile, and edge-computing applications.

## Figures and Tables

**Figure 1 sensors-21-06006-f001:**
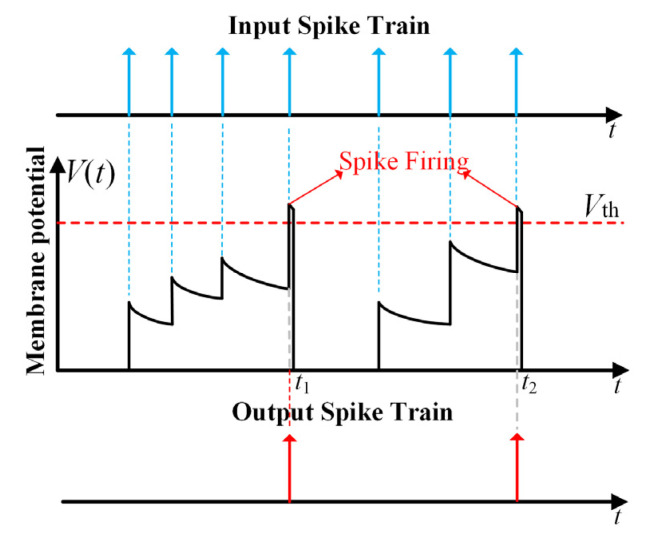
The dynamical behavior of the LIF spiking neuron model.

**Figure 2 sensors-21-06006-f002:**
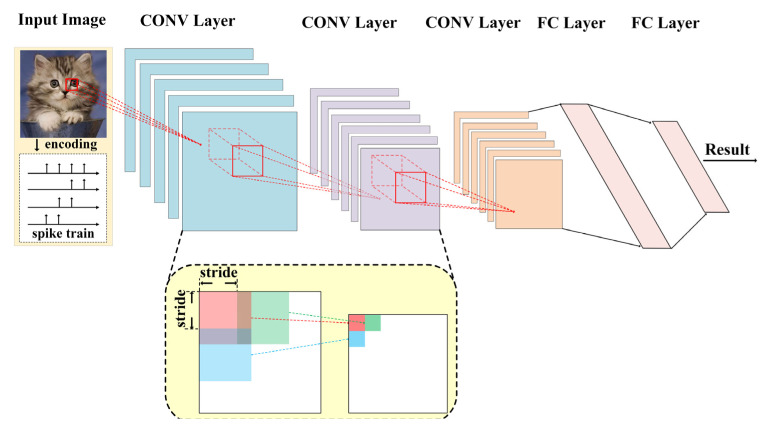
A typical feedforward SCNN topology composed of stacked spiking convolutional (CONV) layers and fully connected (FC) layers.

**Figure 3 sensors-21-06006-f003:**
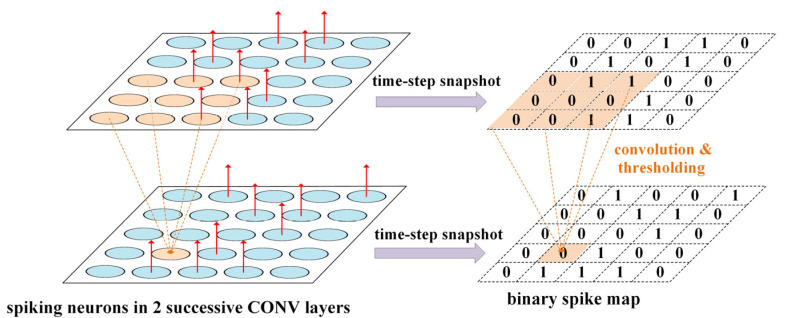
The binary spike maps corresponding to the snapshot post-synaptic spike events in the channels of two successive CONV layers at one time-step. The red arrows indicate the firing neurons at this time-step. We suppose there is only one channel in each layer for simplicity of the illustration.

**Figure 4 sensors-21-06006-f004:**
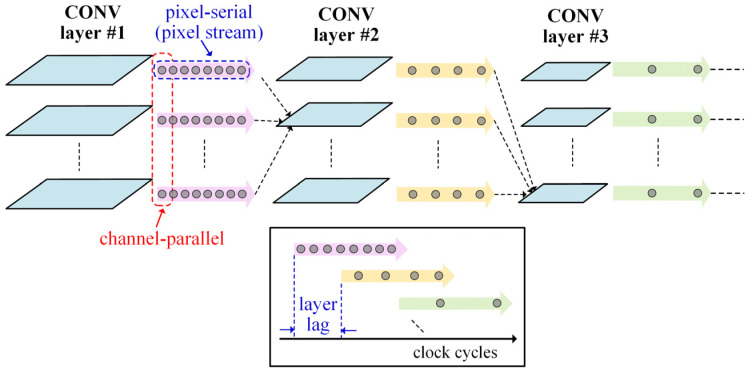
The binary spike map pixel stream processing mechanism for the time-step CNN-like processing in SCNN hardware.

**Figure 5 sensors-21-06006-f005:**
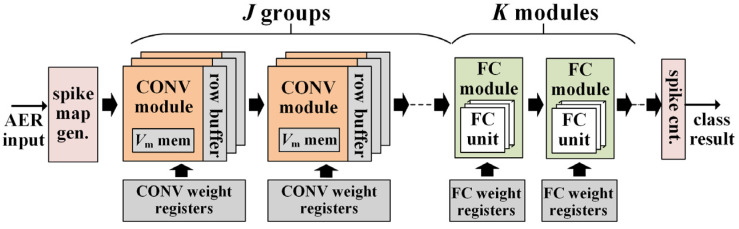
Hardware architecture of the proposed VLSI system.

**Figure 6 sensors-21-06006-f006:**
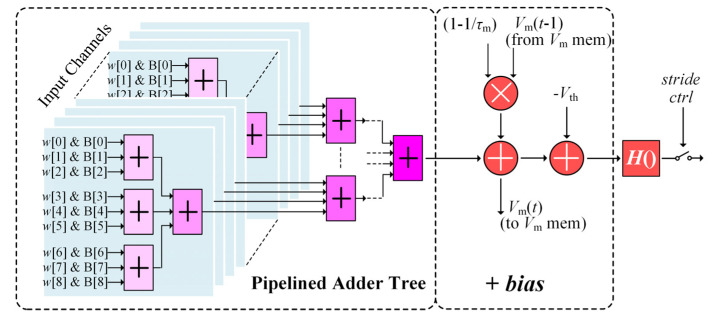
The circuit design of the CONV module with the kernel size of 3 × 3 to process the pixel data from the row buffer.

**Figure 7 sensors-21-06006-f007:**
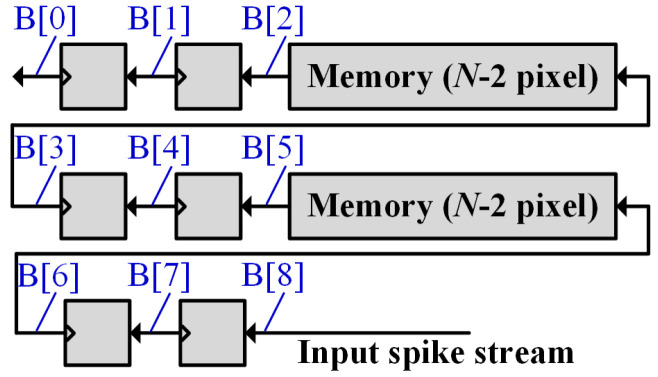
The circuit design of the binary pixel row buffer.

**Figure 8 sensors-21-06006-f008:**
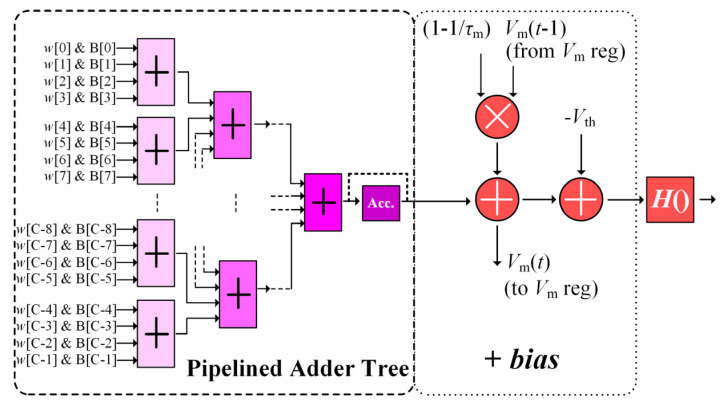
The circuit design of the FC unit.

**Figure 9 sensors-21-06006-f009:**
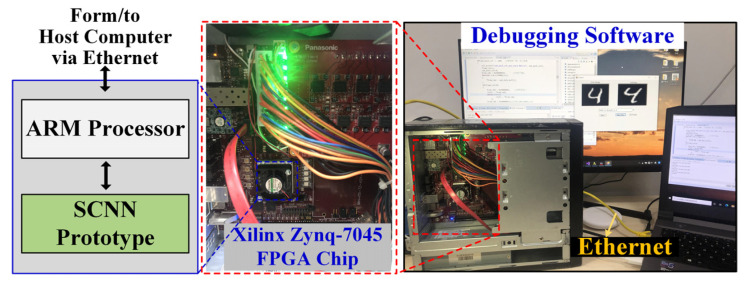
The FPGA prototype and the evaluation system.

**Figure 10 sensors-21-06006-f010:**
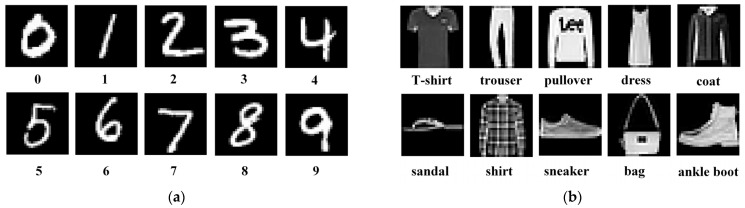
The testing image datasets. (**a**) MNIST. (**b**) Fashion-MNIST.

**Figure 11 sensors-21-06006-f011:**
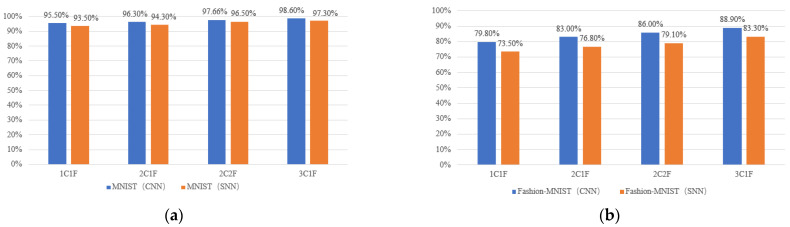
The comparison of recognition accuracies between topologically equivalent non-spiking CNN (computer software) and SCNN (FPGA hardware) on (**a**) MNIST and (**b**) Fashion-MNIST image datasets.

**Table 1 sensors-21-06006-t001:** SCNN model configurations for FPGA prototyping.

Configuration	SCNN Model Structure
1C1F ^1^	16c2-10
2C1F	16c2-32c2-10
2C2F	6c2-32c32-32-10
3C1F	16c1-16c2-32c2-10

^1^*x*C*y*F means *x* CONV layers + *y* FC layers, and *x*c*y* represents a 3 × 3 COVN layer with *x* channels and a stride of *y*, the digit 10 means an FC layer with 10 neurons.

**Table 2 sensors-21-06006-t002:** Resource and power consumptions of our FPGA prototype.

Configuration	Logic Resource			Memory Resource		Power Consumption ^1^
	LUT as Logic (218,600)	FF (437,200)	DSP (900)	Block RAM (545)	LUT as Mem (70,400)	
1C1F	8904 (4.07%)	10,269 (2.35%)	26 (2.89%)	88 (16.15%)	64 (0.09%)	0.519 W
2C1F	64,640 (29.57%)	102,982 (23.55%)	58 (6.44%)	24 (4.40%)	4960 (7.05%)	0.959 W
2C2F	93,202 (42.63%)	136,882 (31.31%)	90 (10.00%)	26 (4.77%)	6123 (8.70%)	1.168 W
3C1F	87,172 (39.88%)	147,832 (33.81%)	74 (8.22%)	32 (5.87%)	6000 (8.52%)	1.241 W

^1^ Estimated by the Xilinx Vavido tool.

**Table 3 sensors-21-06006-t003:** Performance Comparison to software based SNN/SCNN on MNIST.

Ref.	Time (s)	Accuracy (%)
[19]	8649	95.01
[40]	7825	78.5
[41]	5000	99.1
**Ours (3C1F)**	**7.84**	97.3

**Table 4 sensors-21-06006-t004:** Comparison with other SNN/SCNN hardware implementation.

Ref.	Implementation	Clock Freq. (MHz)	Power (mW)	Frame Rate (fps)	Model	Benchmark	Accuracy (%)
[8]	ASIC	75	0.48	N/A	SNN	MNIST	84.5
[9]	ASIC	105	0.16	160	SNN	MNIST	89
[10]	ASIC	25	21	6.25	SNN	MNIST	93.8
[25]	ASIC	100	200	127 ^1^	SCNN	Poker-DVS	N/A
[26]	FPGA	50	0.85	0.4	SCNN	Poker-DVS	96
[27]	FPGA	100	59	111	SCNN	Poker-DVS	N/A
[30]	FPGA	200	N/A	N/A	SCNN	MNIST	99.16
[33]	FPGA	150	4600	164	SCNN	MNIST	98.94
[42]	FPGA	75	1500	6.58	SNN	MNIST	92
**ours** **(3C1F)**	**FPGA**	**100**	**1241**	**1250**	**SCNN**	**MNIST**	**97.3**
						**Fashion-MNIST**	**83.3**

^1^ For the DVS datasets, the frame rate refers to classifications per second on the AER data stream.

## Data Availability

Not applicable.

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
