# Peer review of "A Cost-Efficient High-Speed VLSI Architecture for Spiking Convolutional Neural Network Inference Using Time-Step Binary Spike Maps"

_sensors, 2021, doi:10.3390/s21186006_

Round 1

Reviewer 1 Report

This paper proposes an accelerator for SCNN for embedded scenarios. The paper evaluated the design on FPGA and achieved acceptable accuracy through the MNIST and fashion-MNIST data sets.

1. When comparing the accuracy of MNIST, some references reflecting the state of the art in the field are missing, such as:

Wang, Shu-Quan, et al. "SIES: A Novel Implementation of Spiking Convolutional Neural Network Inference Engine on Field-Programmable Gate Array." Journal of Computer Science and Technology 35 (2020): 475-489.

2. The accuracy of mnist and fashion-mnist has dropped a lot, please explain the reason

3. Please explain how the power consumption measurement is realized. In addition, does it make sense to compare power consumption between FPGA and ASIC?

4. The accuracy of the work in this paper is lower than ref[32]. Is it because the pooling layer is not used?

5. English writing needs improvement.

Author Response

To Reviewer #1:

We greatly appreciate Reviewer’s valuable comments and concerns leading us to significantly improve the quality of our manuscript. We have intensively revised this paper. These major revisions are labeled in yellow sheet background. Line numbers are also appended to the right of the texts on each page. Below are our point-to-point responses to Reviewer’s comments:

  1. When comparing the accuracy of MNIST, some references reflecting the state of the art in the field are missing, such as:

Wang, Shu-Quan, et al. "SIES: A Novel Implementation of Spiking Convolutional Neural Network Inference Engine on Field-Programmable Gate Array." Journal of Computer Science and Technology 35 (2020): 475-489.

We greatly appreciate that Reviewer shares the important literature. We have now added its information to the Introduction (Lines 88-89), and compared with it in Table 4 in Sec 5.2. Some brief discussions are also supplemented (Lines 381-382).

  1. The accuracy of mnist and fashion-mnist has dropped a lot, please explain the reason

Thanks for raising this concern. We started to further tune the algorithm parameters after the initial submission. Now the accuracies on both Fashion-MNIST and MNIST are slightly improved, from 82.3% to 83.3%,and 95.6% to 97.3%, respectively. We have updated the data in Figure 11 (Lines 356-357). The recognition accuracy gap between the two datasets are due to the reason that the Fashion-MNIST object images from the real-world are much more complicated to classification than the handwritten digits in the MNIST dataset. (Lines 342-344) 

  1. Please explain how the power consumption measurement is realized. In addition, does it make sense to compare power consumption between FPGA and ASIC?

We used the Vivado tool to estimate the FPGA power consumption, as already explained in Lines 316-317, We have also added a footnote to indicate it below Table 2 in this revision (Lines 359).

Yes, Reviewer is quite right. It is generally regarded the power consumption of FPGA and ASIC instantiations of the same hardware design cannot be directly compared, as for a particular design, the FPGA chip contains many redundant resources to realize highly flexible functional configuration, and may consume orders of higher power than the ASIC counterpart. We listed the ASIC implementations with the same indices (including the power consumption) only for the information completeness.    

  1. The accuracy of the work in this paper is lower than ref [32]. Is it because the pooling layer is not used?

Thanks for this concern. According to our pilot simulations, the accuracy loss on both MNIST and Fashion-MNIST datasets are only around 0.5 when the pooling layer is replaced by the striding convolution. But the pooling function is very complex in the spike domain and requires many hardware computational resources. So, as a tradeoff, we used the striding convolution to replace the spike pooling layer in our work, as explained in Lines 139 -147.

  1. English writing needs improvement.

Thanks for this comment. We have made our every effort to improve English writing. We would even choose pay to seek commercial assistance on writing after acceptance of this manuscript, if needed.

Thanks again for the Reviewer’s comments on our work.

Reviewer 2 Report

The goal of this paper, as exposed by the authors, is to propose a scalable, cost-efficient and high-speed VLSI architecture to accelerate deep SCNN inference for real-time and low-cost embedded systems. Although the paper is interesting, and the authors have put effort into its elaboration, it has some flaws that make it impossible for me to recommend it for publication. In particular, it is not clear whether the authors' contributions to THIS publication.

Section 1 contain a well-structured introduction and an in-depth review of state of the art.

The aspects related to the real-time system must be explained, presented and validated with practical data (Vivado simulations and oscilloscope capture with stimuli) with special attention because the abstract states "… SCNN inference for real-time low-cost embedded scenarios".

What is the hardware description language used, apart from the on-chip hardware IP core of ARM processor?

Regarding the system design itself (Section 5 in the manuscript), I feel that a lack of details on the exact software tools (Vivado version, simulator or debugger tools, ILA, IPs) and hardware is missing. Optional, FPGA utilization resources, clock timing, power consumption report can also be provided.

Section 5 should continue by adding a chapter of discussions about the tests performed by the authors, as well as the analysis of the contributions made by THIS paper in the context of pixel stream processing, neuromorphic VLSI architectures and SCNN hardware scientific research.

What are the contributions of the authors and the research innovation brought with this publication?

Author Response

To Reviewer #2:

We greatly appreciate Reviewer’s valuable comments and concerns leading us to significantly improve the quality of our manuscript. We have carefully revised this paper. These major revisions are labeled in yellow sheet background. Line numbers are also appended to the right of the texts on each page. Below are our point-to-point responses to Reviewer’s comments:

  1. The aspects related to the real-time system must be explained, presented and validated with practical data (Vivado simulations and oscilloscope capture with stimuli) with special attention because the abstract states "… SCNN inference for real-time low-cost embedded scenarios".

Thanks for raising this concern. we introduced our test platform for the proposed SCNN hardware architecture introduced in Section 5.1 and shown in Figure 9 (Lines 307-311). Unlike analog ASIC design which needs oscilloscope, the timing of digital FPGA implementations is guaranteed by static timing analyzing in the design tool, and in our work, is physically monitored by the ARM IP core with a timer. Such method for real-time performance evaluation is mainly a common trivial engineering trick and we think it is not suitable to be more detailed in an academic research article.   

We have almost re-written the first paragraph of subsection 5.1 (Lines 307-325). We have stated that “the four FPGA prototypes were running under a clock frequency of 100 MHz” (Lines 314-315), and “The prototype inference speed on the MNIST and Fashion-MINIST images for the SCNN model configurations in Table 1 were all as high as around 1250 frame/s (fps), under the 100 MHz clock frequency and with 100 time-steps for each image.” (Lines 339-341)

  1. What is the hardware description language used, apart from the on-chip hardware IP core of ARM processor?

Thanks for reminding us to give this piece of information. We added a sentence at the start of Sec 5.1: “In this work, we used the SystemVerilog to describe the proposed SCNN hardware architecture and the processing circuits...” (Lines 307-311).

  1. Regarding the system design itself (Section 5 in the manuscript), I feel that a lack of details on the exact software tools (Vivado version, simulator or debugger tools, ILA, IPs) and hardware is missing. Optional, FPGA utilization resources, clock timing, power consumption report can also be provided.

Similar to comment #1, the details in using the project design toolkits is a tedious engineering flow which is not proper to be reported in an academic article. The RTL SystemVerilog verification was done in Modelsim, and the synthesis and P&R were simply realized in the Xilin Vivado. The ARM IP core integration employed the IP block design GUI in the Vivado. ILA was not used because we have our own design flow to guarantee the function of the core prototype (mainly in RTL coding verification, a mature framework to correctly integrate the ARM IP core and its software, and the whole evaluation system with our debugging software)… Such engineering flow is so complicated and tedious, and revealing these in a research paper would not contribute to the research innovation.    

Besides, the FPGA utilization resources, clock timing, power consumption have already been reported in our original submission. Please see them in Table 2 and Lines 317 – 318 in this revised submission.

  1. Section 5 should continue by adding a chapter of discussions about the tests performed by the authors, as well as the analysis of the contributions made by THIS paper in the context of pixel stream processing, neuromorphic VLSI architectures and SCNN hardware scientific research.

We greatly thank Reviewer for raising these comments. A new paragraph is now added to the end of Sec 5.2 for discussing these issues. (Lines 383-395)

  1. What are the contributions of the authors and the research innovation brought with this publication?

Thanks. we proposed a scalable, cost-efficient and high-speed VLSI architecture to accelerate deep SCNN inference for real-time and low-cost embedded system. Our SCNN architecture leverages the snapshot of binary spike maps at each time-step along with the spike-map pixel stream processing pipeline to maximize spike throughput, while minimizing the computation and storage consumptions of hardware resources. (Lines 90-98)

The spike map stream processing mechanism proposed by us is introduced in the Sec 3. We explain the working mechanism of spike stream processing technology, and it bring us the following advantages: 1) Small memory footprint; 2) High pixel throughput; 3) Reduced computing resource. Reviewer could also find the summation in the Sec 3. (Lines 196-210)

Thanks again for the Reviewer’s comments on our work.

Round 2

Reviewer 2 Report

The paper was improved by the revision process. I think that the paper can be accepted.